# Revealing a Third Dissolved-Phase Xenon-129 Resonance in Blood Caused by Hemoglobin Glycation

**DOI:** 10.3390/ijms241411311

**Published:** 2023-07-11

**Authors:** Lutosława Mikowska, Vira Grynko, Yurii Shepelytskyi, Iullian C. Ruset, Joseph Deschamps, Hannah Aalto, Marta Targosz-Korecka, Dilip Balamore, Hubert Harańczyk, Mitchell S. Albert

**Affiliations:** 1Faculty of Physics, Astronomy, and Applied Computer Science, Jagiellonian University, 30-348 Krakow, Poland; lutoslawa.mikowska@doctoral.uj.edu.pl (L.M.);; 2Chemistry and Material Science Program, Lakehead University, Thunder Bay, ON P7B 5E1, Canada; vgrynko@lakeheadu.ca; 3Thunder Bay Regional Health Research Institute, Thunder Bay, ON P7B 7A5, Canada; yshepely@lakeheadu.ca; 4Chemistry Department, Lakehead University, Thunder Bay, ON P7B 5E1, Canada; 5Xemed LLC, Durham, NH 03824, USA; iulian@xemed.com; 6Applied Life Sciences Program, Lakehead University, Thunder Bay, ON P7B 5E1, Canadahaaalto@lakeheadu.ca (H.A.); 7Department of Engineering, Physics and Technology, Nassau Community College, New York, NY 11530, USA; 8Faculty of Medical Sciences, Northern Ontario School of Medicine University, Thunder Bay, ON P3E 2C6, Canada

**Keywords:** hyperpolarized ^129^Xe, magnetic resonance spectroscopy, glycation, glucose, hemoglobin

## Abstract

Hyperpolarized (HP) xenon-129 (^129^Xe), when dissolved in blood, has two NMR resonances: one in red blood cells (RBC) and one in plasma. The impact of numerous blood components on these resonances, however, has not yet been investigated. This study evaluates the effects of elevated glucose levels on the chemical shift (CS) and $T_{2}^{*}$ relaxation times of HP ^129^Xe dissolved in sterile citrated sheep blood for the first time. HP ^129^Xe was mixed with sheep blood samples premixed with a stock glucose solution using a liquid–gas exchange module. Magnetic resonance spectroscopy was performed on a 3T clinical MRI scanner using a custom-built quadrature dual-tuned ^129^Xe/^1^H coil. We observed an additional resonance for the RBCs (^129^Xe-RBC1) for the increased glucose levels. The CS of ^129^Xe-RBC1 and ^129^Xe-plasma peaks did not change with glucose levels, while the CS of ^129^Xe-RBC2 (original RBC resonance) increased linearly at a rate of 0.015 ± 0.002 ppm/mM with glucose level. ^129^Xe-RBC1 $T_{2}^{*}$ values increased nonlinearly from 1.58 ± 0.24 ms to 2.67 ± 0.40 ms. As a result of the increased glucose levels in blood samples, the novel additional HP ^129^Xe dissolved phase resonance was observed in blood and attributed to the ^129^Xe bound to glycated hemoglobin (HbA_1c_).

## 1. Introduction

Since its invention [1], hyperpolarized (HP) xenon-129 (^129^Xe) MRI has been primarily used for the functional imaging of the human lungs and for studying pulmonary disorders [2,3,4,5,6,7]. Over the past decade, the feasibility of HP ^129^Xe dissolved-phase imaging has been demonstrated for imaging pulmonary gas transfer [8,9,10,11,12], brain imaging [13,14,15,16,17,18], and kidney imaging [19,20]. HP ^129^Xe dissolved-phase imaging relies on the sufficiently long T_1_ relaxation time in blood (~3.4–7.8 s) [21,22,23,24,25] and the well-distinguished resonances of HP ^129^Xe nuclei dissolved in blood and various tissue compartments. Therefore, the chemical composition of blood should have a critical effect on HP ^129^Xe dissolved-phase imaging since it affects the T_1_ and chemical shift (CS) of the dissolved ^129^Xe resonances. A number of studies have previously evaluated the role of blood oxygenation and its effects on HP ^129^Xe MR spectra [21,22,24,26]. However, there is another vital component of blood chemical composition that has yet to be explored—the glucose level.

A high glucose concentration in blood leads to excessive non-enzymatic chemical interactions between glucose and proteins in the blood [27]. Hemoglobin is the protein most affected by glycation. Structurally, hemoglobin has four subunits, consisting of two α and two β chains, each of which carry an iron-containing heme group responsible for oxygen binding [28]. During the glycation process, a non-enzymatic reaction occurs between glucose and the α-amino groups of valine residues at the N-terminus of the β-chains [29]. The most abundant form of glycated hemoglobin, which is widely used in clinical practice for the evaluation of glycemia in diabetes mellitus, is hemoglobin A1c (HbA1c) [30]. Glycated hemoglobin is naturally present in healthy individuals (~4%), whereas glycation levels up to 20% have been reported in patients diagnosed with diabetes [31]. Elevated HbA1c levels (>6.5%) are routinely used for the diagnosis of diabetes mellitus [32]. The standard HbA1c test is used for the diagnosis of type 2 diabetes and prediabetes. It shows the amount of glycated hemoglobin and reflects the average blood glucose level over the past three months [33].

Hemoglobin glycation induces the formation of oxygen-derived free radicals, which are responsible for causing oxidative stress in erythrocytes [34], leading to an increase in membrane lipid peroxidation [35] and membrane damage in diverse cell types. The glycation of iron-containing heme proteins can cause the degradation of heme and further reactions with H_2_O_2_, leading to increased iron release, as well as ferryl myoglobin formation [27,36]. In addition, iron overload caused by high concentrations of glycated hemoglobin has been shown to be associated with changes in the structure of the red blood cells (RBCs) and increased thrombotic events [37]. Numerous studies have been conducted evaluating the effect of elevated levels of glycated hemoglobin on red blood cell distribution width (RDW); however, the data published has been inconclusive [38]. Some studies have also suggested that hyperglycemia may have some effect on erythropoiesis and RBC survival at concentrations of glucose higher than 40 mM [35,38].

It is known that the HP ^129^Xe signal from blood originates from Xe in plasma and the RBCs [1]. ^129^Xe binds to hemoglobin in hydrophobic cavities close to the external surface in both α and β chains [39]. Therefore, alternations in the structure of hemoglobin caused by glycation are expected to have a direct effect on the HP ^129^Xe magnetic environment and its dipole–dipole interaction with hemoglobin, potentially resulting in CS changes. Furthermore, the overall increase in free radical levels can also be anticipated to cause an effect on HP ^129^Xe CS. The destruction of heme groups accompanied by the release of iron would be expected to have potential effects on HP ^129^Xe-RBC CS, as well as on the $T_{2}^{*}$ relaxation process.

The present lung studies with HP ^129^Xe exploit the measuring of the signal intensity of ^129^Xe bound to tissue and RBC [2,3,4,7,8]. However, the elevated glucose level in blood may have a considerable impact on the ^129^Xe spectroscopic properties and should be taken into account for patients with diabetes.

This work, for the first time, investigates the effect of glucose concentration in blood on the physical properties of dissolved HP ^129^Xe, such as CS and on the effective spin–spin relaxation time $T_{2}^{*}$. Moreover, using non-linear curve fitting, we demonstrated that the spectrum of HP ^129^Xe in blood contains not two (as was conventionally thought) but three dissolved-phase resonances. We report the resonance frequencies, $T_{2}^{*}$ relaxation times, and the CS dependance on glucose level of the three observed spectral components in sterile citrated sheep blood. In addition, we propose a potential mechanism for glycation-related changes in the physical parameters of the dissolved HP ^129^Xe. This research suggests that for a comprehensive understanding of dissolved-phase imaging, glucose levels in the blood should also be taken into account along with blood oxygenation, since both play distinct roles.

## 2. Results

### 2.1. CS Analysis Using a Conventional Three-Peak Model (3PM)

The measurements of the CS of the ^129^Xe dissolved in plasma (^129^Xe-plasma) and ^129^Xe dissolved in RBC (^129^Xe-RBC) resonances were conducted for the glucose concentration range of 0–55 mM. The acquired MRS spectra were first fitted using the conventional three-peak model (3PM): gas peak, plasma peak, and RBC peak. Figure 1A,B show MRS acquired for 10 mM and 45 mM glucose concentrations in the blood, respectively. The increase in glucose concentration did not cause a significant change in the ^129^Xe-plasma CS. Conversely, the ^129^Xe-RBC resonance shifted from 217.86 ppm (10 mM of glucose) to 219.07 ppm (45 mM). Figure 1C shows three representative cumulative Lorentzian 3PM fits for 10 mM, 20 mM, and 45 mM glucose concentrations. The downfield shift of the HP ^129^Xe-RBC resonance with respect to the gaseous ^129^Xe resonance can be clearly observed with increasing glucose concentration. The observed CS increased linearly with the increased glucose concentration (Figure 1D). The fit of the CS dependence of the ^129^Xe-RBC peak on glucose concentration revealed a (0.025 ± 0.004) ppm/mM ^129^Xe-RBC change rate. A strong positive correlation (r = 0.91) was observed between the ^129^Xe-RBC peak CS and the glucose concentration in the measured blood samples. It can be clearly seen that the ^129^Xe-plasma peak was completely unaffected by an increase in glucose concentration (r = 0.1 with a slope of the line equal to (0.001 ± 0.005) ppm/mM. The observed downfield shift of the HP ^129^Xe-RBC resonance is similar to the HP ^129^Xe-RBC resonance frequency change due to the increase in blood oxygenation, albeit with a total span of only ~1.25 ppm.

### 2.2. $T_{2}^{*}$ Relaxation Measurements

Following the measurements of CS, $T_{2}^{*}$ relaxation was assessed based on Lorentzian fits of the measured spectra to 3PM. No significant change was observed in $T_{2}^{*}$ values for HP ^129^Xe dissolved in both RBC and plasma pools with an increase in glucose content (Figure 2). The mean $T_{2}^{*}$ values were equal to (1.48 ± 0.09) ms and (0.87 ± 0.07) ms for HP ^129^Xe dissolved in plasma and RBC, respectively.

### 2.3. Four-Peak Spectroscopic Model (4PM) Analysis

With the blood glucose level increase, the RBC resonance in the ^129^Xe spectrum became more asymmetrical and its linewidth increased slightly (Figure 3). In addition, a small splitting of the RBC peak was noted for higher glucose concentrations (red arrow on Figure 3). Considering the glycation process in RBCs, the observed asymmetry and splitting in the RBC peak can be interpreted by a four-peak spectroscopic model (4PM). In general, structural and functional changes to hemoglobin occur as a result of glycation [27]. Therefore, we can subdivide the RBC pool into HP ^129^Xe bound to non-glycated HbA_0_ and HP ^129^Xe bound to glycated HbA_1c_. These two pools can be characterized by their own HP ^129^Xe resonances. The suggested 4PM includes one gas phase resonance and three dissolved-phase resonances: ^129^Xe-plasma, ^129^Xe-RBC1, and ^129^Xe-RBC2.

The suggested 4PM was used to fit the experimental data (Figure 4A,B). The application of the 4PM did not affect the plasma peak position. The position of the HP ^129^Xe-RBC1 peak was barely affected by glucose: 216.08 ppm for the 10 mM solution and 215.92 ppm for the 45 mM solution. The ^129^Xe-RBC2 peak, however, shifted downfield with increasing glucose concentrations: 220.32 ppm for the 10 mM solution and 220.78 ppm for the 45 mM. The residual error for the 3PM, once plotted, showed distinct peaks at 216, 218, and 222 ppm for the 10 mM sample and at 216 ppm and 219 ppm for the 45 mM sample (Figure 4C,D). Once the spectrum is fitted to the 4PM, however, the residual error becomes flat at the RBC resonance position (between 210 ppm and 240 ppm). This indicates that 4PM fits the acquired HP ^129^Xe blood spectra more accurately compared to the conventional 3PM.

It should be noted that the 4PM fit worked better for glucose concentrations above 5 mM. This was indicated by slightly higher R^2^ values. In addition, small alterations in the peak position for the RBC resonances did not affect R^2^ significantly. Therefore, a lower accuracy of the recalculated spectral parameters can be anticipated for the pure blood and 5 mM samples compared to the samples with higher glucose levels. This is in accordance with our hypothesis that the second HP ^129^Xe RBC peak originates from HbA1c. There is no naturally occurring HbA1c in the pure blood and the concentration is expected to be low in a 5 mM glucose sample.

The proposed 4PM did not affect the CS of the HP ^129^Xe gas peak and ^129^Xe-plasma peak. The HP ^129^Xe-RBC1 CS change was not observed with a glucose level increase (Figure 5A). The HP ^129^Xe-RBC2 resonance frequency, however, increased with a rate of (0.015 ± 0.002) ppm/mM. A strong Pearson correlation was observed between the HP ^129^Xe-RBC2 peak position and the blood glucose level (r = 0.95). The $T_{2}^{*}$ relaxation time for ^129^Xe-plasma did not depend on the glucose concentration in the sample (Figure 5B). Conversely, the ^129^Xe-RBC1 $T_{2}^{*}$ time increased non-linearly from (1.58 ± 0.24) ms up to (2.67 ± 0.40) ms over the studied range of glucose concentrations. The ^129^Xe-RBC2 $T_{2}^{*}$ relaxation time increased from (0.66 ± 0.10) ms to (1.23 ± 0.19) ms over a 0–10 mM glucose concentration range and leveled out at approximately (0.91 ± 0.03) ms at higher glucose levels.

## 3. Discussion

In normal physiological conditions, 95–98% of the hemoglobin in RBCs is present in the HbA_0_ state [31]. Once glucose levels elevate in the blood, HbA_1c_ is generated as part of the glycation process [40]. In fact, the concentration of HbA_1c_ hemoglobin increases linearly with the glucose concentration [30]. Watala et al. demonstrated that there are three major sites where structural changes occur in HbA_1c_: at the amino N-terminus of the β chain (~1/3 of the total amount of glucose binds to this site), α amino sites (~1/3 of the total glycation), and lysine residues (~40% of all glycation) [41]. Even if glucose is mostly linked to N-terminal valine, the glycation affects the spatial structure of the whole hemoglobin molecule. Ye at al. demonstrated that glycation transforms alpha-helices into beta-sheets, which results in conformation changes or the unfolding or even aggregation of hemoglobin [42]. Moreover, Sen at al. observed that the conformation caused by glycation increased the exposure of the hydrophobic tryptophan residues, like Trp14 in alpha-chains, which created one of the xenon-binding sites [27]. These structural changes result in significant alterations in hemoglobin conformation [27,41]. Glycation-induced conformational changes in hemoglobin weaken the heme-globing linkage in HbA_1c_ and make it more thermolabile compared to HbA_0_ [27]. Thus, the global changes in spatial hemoglobin’s structure (tertiary and quaternary structure) may affect the shape of hydrophobic cavities or the binding sites of xenon, potentially resulting in a different chemical shift.

Savino et al. identified a total of twelve ^129^Xe binding sites per Hb_4_ tetramer: the α_1_ chain contains four binding sites; the α_2_ chain contains three; three more are located in the β_2_ chain; and two can be found in the β_1_ chain [43]. Considering the glycation-induced structural changes in HbA_1c_, it is possible and fairly likely that the number, size, and/or spatial location of ^129^Xe binding sites are different in HbA_1c_ compared with HbA_0_. In addition, the glycation of hemoglobin results in the accumulation of free OH∙ radicals due to an iron-mediated Fenton’s reaction [27]. Increasing OH∙ concentrations are anticipated to result in the deshielding of the HP ^129^Xe nuclei. This hypothesis is supported by our experimental data, indicating the linear shift of the RBC HP ^129^Xe resonance downfield with respect to the resonance frequency of the gaseous HP ^129^Xe due to the increase in glucose concentration. The linear change in the HP ^129^Xe CS as a function of glucose level is in agreement with a known linear increase in HbA_1c_ concentration caused by an increase in glucose concentration.

Considering our hypothesis that HP ^129^Xe binding sites are affected by glycation and are different in HbA_1c_ compared to HbA_0_, it is reasonable to assume the existence of two HP ^129^Xe resonances—one from ^129^Xe bound to HbA_0_ and another originating from ^129^Xe bound to HbA_1c_. Therefore, instead of utilizing a conventional three peak model (gas phase resonance, plasma resonance, and RBC resonance) for spectroscopic data analysis, a four-peak model (gas phase resonance, plasma resonance, and two RBC resonances) can be used, according to our hypothesis. Spectral data were analyzed using both models and the residual analyses better supported the hypothesis of the 4PM. After 4PM Lorentzian fits, the residual error was smooth and almost completely flat in the region between 213 and 235 ppm. On the contrary, several residual peaks were observed in the same region after 3PM Lorentzian fits, indicating the presence of certain spectral component that remained unaccounted for by the 3PM. The residual analysis findings were further supported by the visual observation of asymmetry of the RBC peak and the presence of a small peak splitting for higher glucose concentrations.

It should also be mentioned that the 4PM Lorentzian fits of the spectroscopy data acquired for higher glucose concentrations were much better compared to the 4PM Lorentzian fits for the low glucose levels. For the pure blood samples as well as for the 5 mM glucose concentration, perturbations in ^129^Xe-RBC1 and ^129^Xe-RBC2 peak positions did not affect R^2^ significantly, if at all. Therefore, it can be concluded that the 4PM functions better for blood glucose levels above 5 mM. For concentration ranges between 0 mM and 5 mM, the 4PM can potentially result in some level of inaccuracy due to the four-peak fitting process. This can plausibly be explained by a low glycation level of the 0 mM and 5 mM samples.

The utilization of the 4PM demonstrated that the CS of one of HP ^129^Xe-hemoglobin peaks (RBC1) is independent of glucose level, whereas the resonances of the RBC2 peak experience a downfield shift linearly with an increase in the glucose level, and thereby, an increase in HbA_1c_ concentration. Although the 4PM preserves the overall linear trend of the ^129^Xe CS evolution, the net span of CS changes was reduced from ~1.25 ppm (for 3PM) down to ~0.9 ppm. The CS change in HP ^129^Xe resonances due to the glycation of hemoglobin is much smaller than the previously reported changes due to the blood oxygenation by Norquay et al. [24]. It should be mentioned that the actual physiological range of glucose levels in the blood has an upper limit of ~35 mM. Therefore, the net expected physiologically relevant change in HP ^129^Xe RBC resonances due to HbA_1c_ formation is limited to ~ (0.53 ± 0.07) ppm, which is one order of magnitude smaller compared to the blood oxygenation CS changes. The plasma resonance position was unaffected by the elevation of the blood glucose level.

The analysis of the $T_{2}^{*}$ relaxation times indicated that the HP ^129^Xe-plasma resonance was unaffected by glucose level. Interestingly, the $T_{2}^{*}$ values for the RBC1 peak increased non-linearly over the studied range of glucose concentrations. The $T_{2}^{*}$ dynamics of the RBC2 peak were similar for glucose levels below 10 mM, albeit leveling out at higher concentrations.

Considering that the HP ^129^Xe-plasma resonance was completely unaffected by alternations in glucose concentration, the detected changes of HP ^129^Xe RBC relaxation should originate from internal alternations in RBCs’ magnetic susceptibilities and/or from changes in HP ^129^Xe spin–spin interactions with hemoglobin. There are two plausible mechanisms accounting for HP ^129^Xe transverse relaxation changes within the RBC. It was previously suggested that glycation induces iron release from heme pockets of hemoglobin [27] and, therefore, should reduce the spin–spin interaction between HP ^129^Xe enclosed in hemoglobin and the heme–iron atoms. Alternatively, conformational changes of HbA_1c_ may occur in a way that the heme pockets become spatially distanced from HP ^129^Xe binding sites, thereby reducing the HP ^129^Xe—namely iron spin–spin interactions. Considering that iron release and conformational changes occur during the glycation process itself, it is possible that both mechanisms contribute to the observed changes in the $T_{2}^{*}$ relaxation of HP ^129^Xe within the RBC.

Although a careful study of the position of the ^129^Xe binding sites in HbA_1c_ is required for a proper explanation of the observed results, it is plausible that conformational changes in hemoglobin due to glycation occur in such a way that the dissociation constant of the HP ^129^Xe-HbA_1c_ becomes substantially lower compared to the dissociation constant of the HP ^129^Xe-HbA_0_ complexes. Unfortunately, an accurate assessment of the relative changes in dipole–dipole interaction strength is not possible without knowledge of the location of ^129^Xe atoms in HbA_1c_.

Despite the fact that the knowledge of the ^129^Xe-HbA_1c_ binding sites is of critical importance before attempting a quantitative description of the relaxation mechanisms responsible for the changes in HP ^129^Xe relaxation and the alternations of the CS, it should be possible to hypothesize the origin of the ^129^Xe-RBC1 and ^129^Xe-RBC2 peaks by combining our experimental results with previous knowledge on the glycation process and HbA_1c_. Due to the glycation process, iron is released from the heme pocket of hemoglobin [27]. In addition, hydrogen peroxide is produced by autoxidizing glucose, and its concentration builds up with time [44]. Hydrogen peroxide further initiates iron release from hemoglobin [45], affecting HbA_1c_ much faster compared to HbA_0_ [27]. The iron released from heme pockets participates in an iron-dependent Fenton’s reaction that becomes a source of free OH∙ radicals within RBCs [27]. The charge of free radicals deshields the HP ^129^Xe nuclei, resulting in a downfield chemical shift detected via MRS. On the other hand, the release of iron may result in a decrease in the spin–spin interaction between the HP ^129^Xe bound to the hemoglobin. Since the iron release is much higher for HbA_1c_, it is plausible that the reduction in spin–spin interaction would be substantial for HP ^129^Xe enclosed within HbA_1c_. The $T_{2}^{*}$ of HP ^129^Xe bond to HbA_0_ would be less affected due to the lower amount of iron released. Therefore, based on our results, we hypothesize that the RBC1 peak originates from HP ^129^Xe-HbA_1c_, while the RBC2 peak corresponds to the HP ^129^Xe-HbA_0_ signal. It is further plausible that the CS of the HP ^129^Xe-HbA1c signal is unaffected by the free OH∙ radical levels due to the conformational changes of the HbA1c and potential enclosure of HP ^129^Xe within the protein structure. On the other hand, the HP ^129^Xe-HbA_0_ signal experiences a linear downfield CS change due to the interaction with free radicals. It is also possible that the small change in $T_{2}^{*}$ of HP ^129^Xe-HbA_0_ is a result of substantially slower iron extraction from HbA_0_.

Our study has several limitations. First, the initial incubation time of the blood samples with glucose solution was relatively short (only ~1 h). The glycation process is relatively slow, consisting of several steps and lasting a span of several days [44]. In the first step, the non-enzymatic Maillard early phase reaction takes place. This occurs after the binding of glucose transferred from the blood to the hemoglobin in RBCs and dictated by the interaction between the aldehyde group of the reducing glucose with the N-terminal amino base and ε-amino base of the lysine residue with the consequent formation of the Schiff base (almidine). This reaction is reversible—it is followed by an irreversible Amadori rearrangement, which results in the production of a stable form of HbA_1c_ (ketoamine) [46]. The Amadori rearrangement is considered to be the limiting reaction in the glycation process [47,48]. The amount of HbA_1c_ and synthetized H_2_O_2_, and therefore, the final concentration of free radicals, increases gradually over the glycation time course. Koga et al. used a 1 h incubation time in their study and confirmed the formation of the labile-glycated hemoglobin (aldimine); however, the formation of stable glycated hemoglobin (ketoamine) was not observed [46]. Thus, it is likely that we observed only the initial changes in the HP ^129^Xe CS and $T_{2}^{*}$ relaxation time. A longer incubation time period will result in more pronounced changes in CS and in $T_{2}^{*}$ relaxation time for HP ^129^Xe dissolved in RBCs. It should be noted that longer incubation times must be moderated by the fact that a small percentage of hemoglobin will be converted into methemoglobin in standing blood over time [49]. The formation of methemoglobin from hemoglobin within the red blood cells is an ongoing oxidative process that can result from the exposure of blood to the atmosphere in long standing blood samples. Methemoglobin forms when hemoglobin is oxidized to contain iron in the ferric [Fe^3+^] state rather than the normal ferrous [Fe^2+^] state [49]. Since methemoglobin is paramagnetic, this itself will cause additional changes to the CS and $T_{2}^{*}$ relaxation time [23]. Therefore, longer incubation times are a tradeoff between allowing the full glycation of the blood samples versus limiting the formation of methemoglobin. In addition, all experiments were conducted on sterile citrated sheep blood. While it is not clear how the presence of citric acid affects protein glycation, the reproduction of these experiments may be performed in other types of animal blood.

Our work is a pioneering proof-of-concept study that should be further expanded by performing experiments with purified human hemoglobin, which is very low in the glycated form and separated RBCs from healthy and hyperglycemic subjects. This will allow us to study the effects of glucose on human RBCs. Consequently, the present study can be translated to the utilization of human blood drawn from healthy participants and hyperglycemic participants to observe how glycose-related changes are affected by the presence of other biological moieties from human blood. Furthermore, glycation level measurements are vital for obtaining reliable conclusions regarding the glucose effect on the ^129^Xe spectroscopic properties in blood. Additionally, no studies have yet been performed that investigate HP ^129^Xe binding to the glycated form of hemoglobin, which can be assessed in future X-ray diffraction (XRD) studies.

## 4. Materials and Methods

### 4.1. Sample Preparation

A 500 mM solution of glucose was prepared by dissolving 3.6 g of D-glucose (Sigma-Aldrich, St. Louis, MO, USA) in 40 mL of 1× phosphate-buffered saline (PBS) at pH 7.4 at room temperature. The mixture was stirred for complete dissolution and used as a stock solution for the preparation of the blood samples. Various volumes of the glucose solution in PBS were mixed with 20 mL of fresh citrated sheep blood (Cedarlane, Burlington, CA, USA) to create the following set of concentrations: 5, 10, 15, 20, 25, 35, 45, and 55 mM. The volume of glucose solution added to the blood was at least one order of magnitude smaller than the volume of the blood, in order to minimize blood dilution effects. A total of 20 mL of the pure citrated sheep blood without any additives served as a control sample. All blood samples were allowed to equilibrate to room temperature for approximately 1 h. The reported incubation times for glycation in the literature vary between 30 min up to hundreds of hours, depending on the incubation protocol [50]. Moreover, the vast majority of studies were able to confirm hemoglobin glycation for all of the incubation time periods. Therefore, successful hemoglobin glycation is expected within our experimental time frame.

### 4.2. Magnetic Resonance Spectroscopy (MRS)

^129^Xe gas was polarized up to 56% using a XeBox-10E polarizer (Xemed, Durham, NH, USA) and disposed into 1 L Tedlar bags. The experimental setup (Figure 6) used for mixing HP ^129^Xe with blood was similar to that used by Norquay et al. [25]. Mixing was performed using an exchange module (Superphobic MicroModule 0.5 × 1 G680 Contactor; Membrana, Charlotte, NC, USA). A steady flow of HP ^129^Xe was set through the exchange module with the help of a pressure chamber pressurized with a continuous flow of N_2_ gas. The flow rate of N_2_ into the pressure chamber was controlled by a ventilator. The solution of sheep blood in the 10 mL syringe was connected to an exchange module and placed in a custom-built dual ^1^H/^129^Xe quadrature MRI birdcage coil, while the other empty syringe was connected to the other side of the exchange module. The blood was pumped back and forth manually through the exchange module perpendicular to the ^129^Xe flow for ~6 s. This technique allowed a sufficient amount of HP ^129^Xe to dissolve in the sheep blood and avoid the formation of gas bubbles in the blood.

A clinical Philips Achieva 3.0 T MRI scanner (Philips, Andover, MA, USA) equipped with a custom-built dual ^1^H/^129^Xe quadrature MRI birdcage coil was used for all ^129^Xe-blood spectroscopy acquisitions. The coil was tuned to 35.33 MHz, and the scanner was shimmed on the ^1^H signal from blood to correct for B_1_ inhomogeneities. Prior to taking any measurements, the coil was calibrated for the blood samples by applying a series of ten RF pulses with a s10° flip angle centered on the gas peak position of HP ^129^Xe resonance (for this, 1 mL syringes filled with HP ^129^Xe were placed into the coil near the blood sample).

To measure the effect of glucose concentration on the CS of the ^129^Xe dissolved in plasma (^129^Xe-plasma) and in RBC (^129^Xe-RBC) resonances, high-resolution single voxel spectroscopy was acquired for all samples. The receiver bandwidth was equal to 22 kHz and the number of samples was set to 4096, yielding a 0.15 ppm spectral resolution. A total of 0 ppm was set in between the peaks of ^129^Xe dissolved in plasma and the RBCs. A 90° rectangular excitation pulse was utilized. An FID spectrum was acquired with a TR of 189.6 ms and a TE of 0.25 ms. Measurements were repeated five times for each sample.

### 4.3. Data Reconstruction and Statistical Analysis

All data were initially analyzed using custom MatLab scripts in Matlab 2020b (Mathworks, Natick, MA, USA). The MRS spectra were postprocessed in Origin2021b (OriginLab, Northampton, MA, USA) for CS and $T_{2}^{*}$ relaxation time assessment. HP ^129^Xe MRS spectra were fitted to either three (gas resonance, ^129^Xe-plasma resonance, and ^129^Xe-RBC resonance) or four (gas resonance, ^129^Xe-plasma resonance, and two ^129^Xe-RBC resonances) Lorentzian curves. The quality of fit was accessed via R^2^ value and through the residual fit error, which was automatically calculated at the end of the fitting algorithm. The comparison between the three and four peak model was performed based on the R^2^ and the residuals.

The full width half maximum (FWHM) of Lorentzian curves was utilized for calculations of the $T_{2}^{*}$ of ^129^Xe dissolved in RBCs and plasma using the following equation [51]:(1)$$T_{2}^{*}=\frac{1}{\pi *FWHM}.$$

To evaluate statistical significance of the acquired results, a paired two-tailed *t*-test was used with a statistical significance level of 0.05. Pearson’s correlation coefficient was calculated between the CS changes of each HP ^129^Xe dissolved phase resonance and the glucose concentration. All statistical analysis was conducted using Origin2021b v.9.8.5.212 software.

## 5. Conclusions

In this work, for the first time, we demonstrated glucose-induced changes in CS and $T_{2}^{*}$ relaxation of HP ^129^Xe dissolved in blood. By using a residual analysis, we observed the evidence for a third dissolved-phase resonance, which was attributed to HP ^129^Xe bound to HbA_1c_ produced during HbA_0_ glycation. Our four-peak model suggests that one dissolved ^129^Xe resonance originates from plasma, whereas two other peaks originate from ^129^Xe encapsulated by HbA_0_ and HbA_1c_ (the fourth peak stems from the gas phase resonance). We observed a linear dependence of the ^129^Xe-HbA_0_ resonance CS on the blood glucose levels. The ^129^Xe-HbA_1c_ $T_{2}^{*}$ changed non-linearly with an increase in glucose level. The HP ^129^Xe-plasma resonance was not affected by alternations in the glucose level.

## Figures and Tables

**Figure 1 ijms-24-11311-f001:**
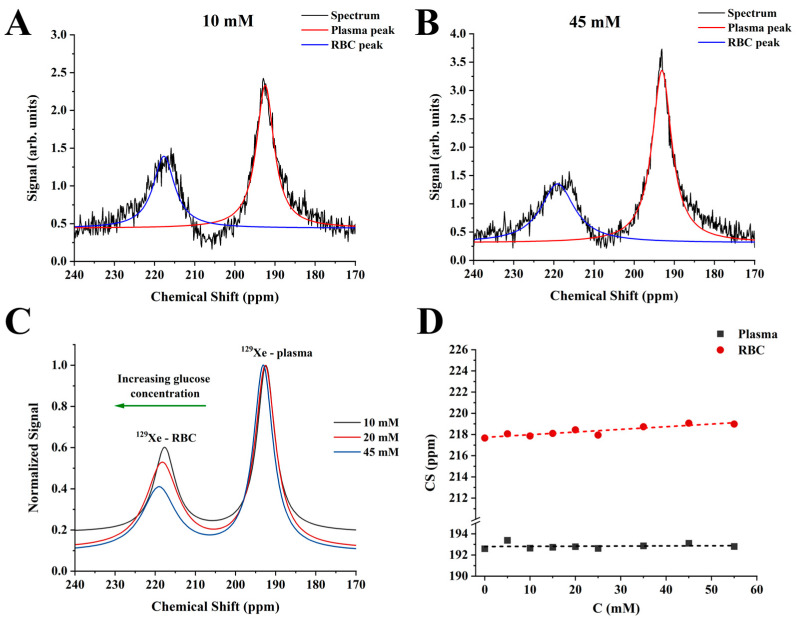
HP ^129^Xe MRS spectra of the 10 mM sample (**A**) and 45 mM sample (**B**) fitted to the conventional 3PM. The CS scale was selected to depict dissolved-phase resonances. The black line corresponds to the acquired HP ^129^Xe spectra. The red and blue lines are Lorentzian fits of the ^129^Xe-plasma and ^129^Xe-RBC peaks, respectively. (**C**) Three fitted HP ^129^Xe MRS spectra of 10 mM, 20 mM, and 45 mM samples demonstrate a noticeable downfield shift of the ^129^Xe-RBC resonance with a glucose concentration increase. The CS versus glucose concentration dependence of HP ^129^Xe-RBC peak appeared to be linear (**D**), whereas the CS of the HP ^129^Xe dissolved in plasma remained unaffected by blood glucose concentration.

**Figure 2 ijms-24-11311-f002:**
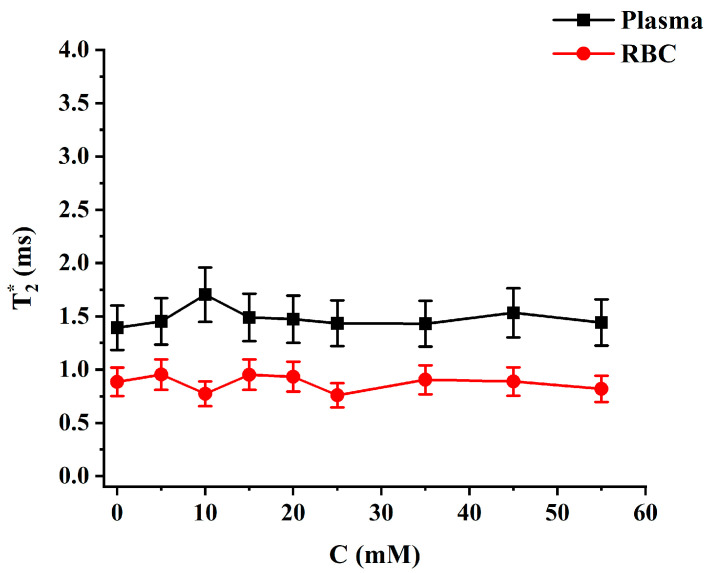
HP ^129^Xe $T_{2}^{*}$ relaxation dependance on blood glucose level for ^129^Xe-plasma (black line) and ^129^Xe-RBC (red line). $T_{2}^{*}$ values were not affected by the blood glucose level.

**Figure 3 ijms-24-11311-f003:**
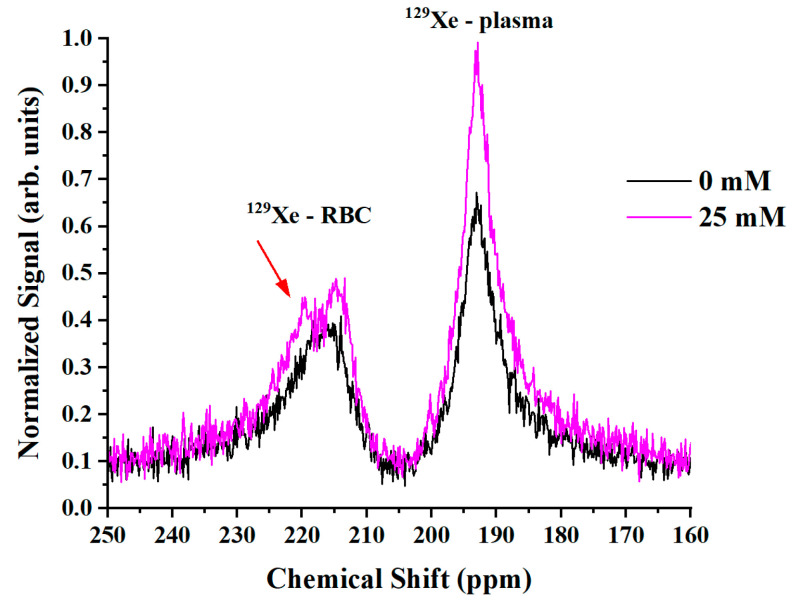
HP ^129^Xe MRS spectra acquired for a pure blood sample (black line) and for a 25 mM blood glucose concentration (purple line). The RBC peak became asymmetrical and broader with the addition of glucose. A small splitting of the RBC peak can be seen (red arrow).

**Figure 4 ijms-24-11311-f004:**
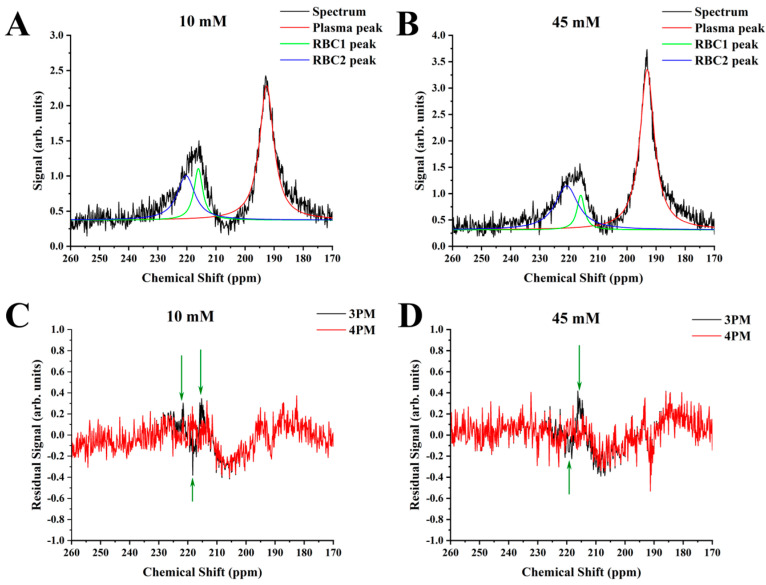
HP ^129^Xe MRS spectra of the 10 mM sample (**A**) and 45 mM sample (**B**) fitted to the proposed 4PM. The CS scale was selected to depict dissolved-phase resonances. The black line corresponds to the acquired HP ^129^Xe spectra. The red line corresponds to the Lorentzian fit of the plasma resonance. The green and blue lines represent the Lorentzian fit of the ^129^Xe-RBC1 and ^129^Xe-RBC2 peaks, respectively. The residuals of 3PM (black lines) and 4PM (red line) were plotted as a function of chemical shift for the 10 mM sample (**C**) and 45 mM sample (**D**). It can be clearly seen that the conventional 3PM shows some residual signal present (green arrows). The proposed 4PM, however, results in a flat baseline with almost no residual signal at the RBC resonance position (between 210 ppm and 240 ppm) indicating a better fit of the RBC signal. Therefore, the proposed 4PM better suits the acquired HP ^129^Xe MRS spectra analysis compared to the conventional 3PM.

**Figure 5 ijms-24-11311-f005:**
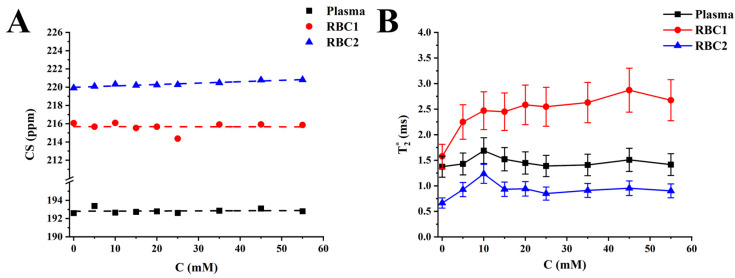
HP ^129^Xe 4PM CS measurements and $T_{2}^{*}$ relaxometry. (**A**) HP ^129^Xe CS dependences for ^129^Xe-plasma (black), ^129^Xe-RBC1 (red), and ^129^Xe-RBC2 (blue) as a function of blood glucose concentration. The dashed lines correspond to a linear fit of the measured CS dependances. A strong positive correlation can be observed between ^129^Xe-RBC2 CS and glucose level, whereas the CS of HP ^129^Xe-plasma and ^129^Xe-RBC1 remains unaffected by blood glucose. (**B**) HP ^129^Xe $T_{2}^{*}$ relaxation dependence on blood glucose level for ^129^Xe-plasma (black line), ^129^Xe-RBC1 (red line), and ^129^Xe-RBC2 (blue line). $T_{2}^{*}$ values of HP ^129^Xe in plasma were not affected by the blood glucose level, whereas the $T_{2}^{*}$ relaxation time of ^129^Xe-RBC1 increased non-linearly with a glucose concentration increase. $T_{2}^{*}$ values of the ^129^Xe-RBC2 peak increased slightly over the range of 0–10 mM and then leveled out.

**Figure 6 ijms-24-11311-f006:**
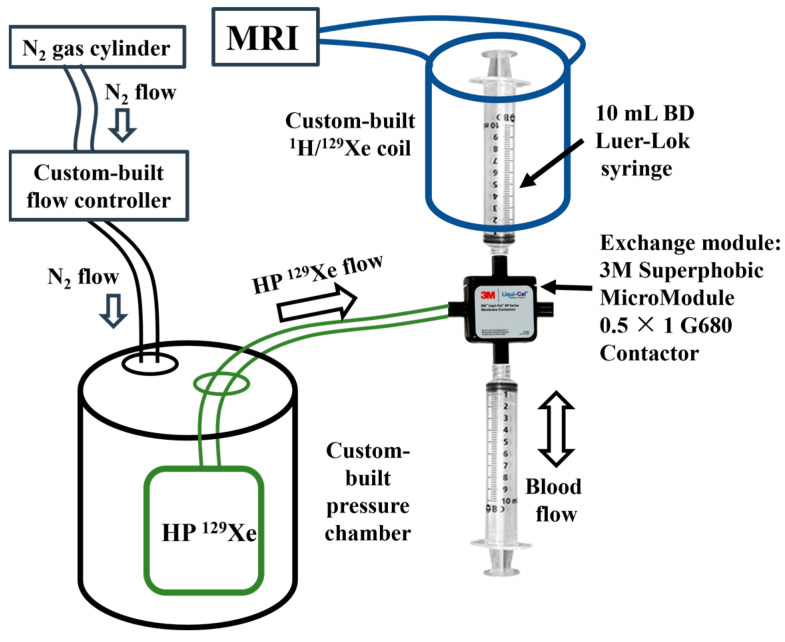
Setup used for mixing the blood with HP ^129^Xe and further MR signal acquisition. A steady flow of HP ^129^Xe from the 1 L bag was regulated by the continuous flow of N_2_ in the pressure chamber, forcing the HP ^129^Xe through the exchange module. The mixing of blood and HP ^129^Xe was performed by manually pumping the blood through the exchange module.

## Data Availability

The data that support the findings of this study are available from the corresponding author upon reasonable request.

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
