# Peer review of "Revealing a Third Dissolved-Phase Xenon-129 Resonance in Blood Caused by Hemoglobin Glycation"

_ijms, 2023, doi:10.3390/ijms241411311_

Round 1

Reviewer 1 Report

Dear Authors,

ijms-2446632

Revealing a third dissolved-phase xenon-129 resonance in blood caused by hemoglobin glycation

I have gone through your manuscript.

Glycation is one of the key factors in the development and progression of various diabetic complications.

  I greet the authors for selecting this research work on "the effect of glucose concentration in blood 82 on the physical properties of dissolved HP xenon-129".

Below are my comments:

1. Line no: 101 to 102: Results will be clearer if you remove the following line, "The black line corresponds.....129Xe-RBC peaks, respectively". Also, you have described the same in the caption of Figure 1.

2.   Line No: 143: 2.3: green arrow on Fig .3 ?

3. Authors can include the blood collection procedure and / or the source details in the materials and methods

4. I suggest that the authors include Ethical statement (committee's approval)

5. Add future directions of this research 

6. There are some formatting errors. Correct them.

7. Mention how your findings can help the medical fraternity.

8. Also, add the prescribed duration with regard to HbA1c testing for diabetes diagnosis in the introduction (average blood glucose measurement.... 90- 120 days..)

The present form is fine.

Reviewer 2 Report

Dear Authors, dear Editor,

draft “ijms-2446632-peer-review-v1 - Revealing a third dissolved-phase xenon …” describes the observation of a separate resonance of 129-Xe in whole sheep blood, the positon (chemical shift) of which drifts downfield with the increasing concentration of blood glucose. They assign the second signal to Xe bound to glycated haemoglobin. They obtain their results by fitting the signals to a four-peak model tht fits better than a three-peak model to the spectral profile.

This is an interesting observation that is worth publication in order that others can verify the same finding and exploit its value.

However, I have a doubt that, given the novelty and importance of the finding, several other interested readers may raise. First, why sheep blood and not human blood, which may have more direct application to clinical experiments? Confirmation experiments can be devised by using purified human Hb, which is very low in the glycated isoform, and in samples of human blood and separated RBC from normal and hyperglycaemic subjects.

At least from the reported information, there is no independent assessment of haemoglobin glycation of sheep blood. I wonder whether glycation is so fast that there is substantial reaction within the time of the experiment, just a few minutes. The Authors should demonstrate Hb glycation with other complementary techniques, such as a mass spectrum with peaks for sheep Hb chains and corresponding glycate isoforms. Next, the question stands whether downshielding is due to solvation from a hydrophobic protein domain or from other molecules. In the cited Savino (2009), Xe atoms are embedded in the hydrophobic core, while the N-terminal valine that carries the additional glycation is outside the core. This makes so many doubts that the Authors should satisfy.

I strongly hope that you may consider my criticism as strong professional appreciation of this initial work and exploit my points to make a more robust point of your findings. I regretfully mark this draft for rejection just in order to encourage you to expand your experiments and make a stronger point of your findings.

Kind regards

little if any

Round 2

Reviewer 2 Report

Dear Authors, dear Editor,

I have read with utmost attention the revised version of the draft and the accompanying response to my points.

While I confirm my perplexity on the contents of this very exploratory work, I believe that you should get it published (and advertised by the Publisher) due to its novelty and the necessity that its findings can be studied with better detail, something that will not be possible without a founding publication. I completely agree with the Authors when they point to the difficulty of being allowed to work with biological samples, such as with human blood, without consent from the Ethical Committee. I hope that publication of this report will help in obtaining the proper authorization and grants to continue this research. Also, best wishes for repair of your instrument!

I strongly suggest that the Authors incorporate their answers to my points in the Discussion or Conclusions, because this will help many interested readers to understand the meaning of this necessarily incomplete pioneer work, without incurring in the same doubts I had myself and demeaning its contents.

Please, feel my deepest compliments and hope that you will be able to go forward with this research.

Kind regards
